# Comparison of Important Quality Characteristics of Some Fungal Disease Resistance/Tolerance Grapes Dried with Energy-Saving Heat Pump Dryer

**Arif Atak** [1,*] , **Zekiye Göksel** [2] , **Cüneyt Tunçkal** [3] **and Yusuf Yılmaz** [4,*]

1 Department of Viticulture, Atatürk Horticultural Central Research Institute, 77102 Yalova, Turkey

2 Department of Food Quality, Atatürk Horticultural Central Research Institute, 77102 Yalova, Turkey; zekiyegoksel@gmail.com

3 Air Conditioning and Refrigeration Program, Department of Electric and Energy, Yalova Community College, Yalova University, 77102 Yalova, Turkey; cuneyt.tunckal@yalova.edu.tr

4 Department of Food Engineering, Faculty of Engineering and Architecture, Burdur Mehmet Akif Ersoy University, 15030 Burdur, Turkey

* Correspondence: atakarif@gmail.com (A.A.); yilmaz4yusuf@yahoo.com (Y.Y.); Tel.: +90-505-4804130 (A.A.)

**Abstract:** Raisins have been widely consumed for many years all around the world, and different grape cultivars and drying techniques are used in their production. Recently, mechanical drying systems have been used to overcome any undesirable effects that arise from sun-drying with grape cultivars that require fewer pesticides to minimize the risk of residues. Both seeded and seedless cultivars were preferred for drying in the past; however, seedless grape cultivars have been increasingly preferred for drying purposes in addition to their use as table grapes. For the first time, an alternative processing method (using an energy-saving heat pump dryer) and important quality characteristics (total phenolics, antioxidant activity, brix, colour analyses and sensory evaluation) of disease resistant/tolerant grape cultivars of different species that can be grown in humid regions were investigated in this study. First, the fresh fruits of nine different grape (*Vitis* spp.) cultivars grown in a humid ecology were analysed, and then so too were the important phytochemical and quality characteristics of raisins dried with an energy-saving heat pump dryer. The water activity of the raisins ranged from 0.71 (Özer Beyazı) to 0.42 (Kay Gray). The total phenolic contents of Muscat Bleu (65.96), Philipp (64.88) and Campbell Early (64.53 g GAE/100 g db) berries were the highest ($p < 0.05$). The fresh berries of the Kishmish Vatkana cultivar had the lowest antioxidant activity, as determined by the 2,2-diphenyl-1-picrylhydrazyl (DPPH) assay (525.81 mmol TE/100 g) ($p < 0.05$). The total phenolic contents of raisins ranged from 12.76 to 13.58 g GAE/100 g fw. The highest value on a dry weight basis was 19.30 g GAE/100 g for the raisins of the Özer Beyazı cultivar ($p < 0.05$). The highest antioxidant activity value on a dry weight basis was found for the raisins of Özer Beyazı (991.01 mmol TE/100 g) using the DPPH assay. The raisins of the Philipp cultivar had an antioxidant activity of 7893.51 mmol TE/100 g db, as determined by the ferric reducing antioxidant potential (FRAP) assay, which was significantly higher than those of other cultivars, with the exception of Muscat Bleu ($p < 0.05$). The range for antioxidant activity values provided by the cupric reducing antioxidant capacity (CUPRAC) assay was high, and the highest value was found for the raisins of the Philipp cultivar (4505.21 mg Trolox/100 g fw) ($p < 0.05$). The results indicated that the raisins of the seedless Rhea and Kishmish Vatkana cultivars can be appreciated more than those of the other cultivars, both in terms of their bioactive content and sensory scores, and the raisins of these cultivars, which can be grown in humid regions, hold a great deal of potential for grape growers.

**Keywords:** phenolics; grapes; *Vitis*; antioxidant activity; sensory analyses

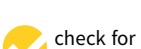



## 1. Introduction

Dried fruits are popular snacks due to their bioactive compounds and nutritional and functional properties [1]. Most of dried fruits are a part of the Mediterranean diet because

of their unique aroma and texture. In particular, raisins (*Vitis* spp.) are among the most popular dried fruits. Raisins are produced by drying fresh grapes and have been known of since the Neolithic period [2].

Fresh grapes have relatively high sugar and moisture contents and are very sensitive to microbial spoilage during storage. Therefore, after harvesting, they must be consumed fresh or processed into various products such as wine, grape juice, jam and raisins in a few weeks in order to reduce economic losses [3].

Drying fresh grape berries into raisins is a major processing method in many grape-producing countries such as Turkey, the US, China, Iran and Uzbekistan, which are the top five countries in terms of raisin production. In these countries, different seeded or seedless grape cultivars are used as sources of raisins [4]. The Aegean region, lying alongside the Aegean Sea, is the most important grape-producing zone in Turkey and mainly produces seedless grapes used in raisin production. Table grapes make up about 51 percent of production, whereas drying grapes make up 38 percent (27 percent for seedless raisins and 10 percent for seeded raisins) and 11 percent is for wine production [5]. Raisins, with their attractive flavour and texture and with them being rich in nutritional content, are extensively consumed around the world as a snack or as a food ingredient in the cooking, baking and brewing industries [6].

The Sultani cultivar is the main seedless grape cultivar used in raisin production in Turkey and many other countries. In addition to Sultani, various seeded or seedless local cultivars have been widely used in the production of raisins [7,8]. Depending on the cultivation conditions, the drying process of grapes varies in different parts of the world. Traditionally raisins are produced via the sun drying of grape berries for 8–10 days, which substantially reduces the water content of fresh grapes. Even though this method is inexpensive, there is a high risk of physical or biological damage to dried products, such as via dust and insect infection. There are three main methods that are conventionally used in raisin production, which are sun drying, shade drying and mechanical drying. Sun drying or solar drying has several disadvantages including the possibility of environmental contamination due to dust and insect infections, physical and/or microbial deterioration caused by rain and colour deterioration due to intense solar radiation. Moreover, the removal of contaminants (e.g., small stones, soil, leaves, dust, etc.) is tedious during the raisin cleaning process [9]. Mechanical drying, which is safe, rapid and controllable in preferred during raisin production when high throughput is needed [10]. Traditional hot air drying is the most common drying method; however, the high energy consumption of this method and the need for high drying temperatures are major drawbacks of this technique. Drying methods such as freeze drying, hot air and oven drying, vacuum drying, microwave and infrared drying all have higher energy requirements. In order to eliminate these disadvantages, studies on alternative technologies have been carried out in recent years. With its low energy consumption, heat pump drying (HPD) technology has been recently developed as a highly preferred drying method. HPD consumes 60–80% less energy than conventional dryers at the same temperatures [11]. Also, another advantage of HPD is that the moisture content and drying air temperature can be easily controlled. For this reason, it was used in this study as it is a very suitable method for drying many heat- and moisture-sensitive products [12].

With an increase in environmental and human health awareness in recent years, the interest in grape cultivars that are resistant or tolerant to diseases, which means the production of healthy products with fewer pesticides, has increased [13]. On the other hand, the fresh consumption of these grapes is somewhat limited in terms of consumer acceptance since they generally have a hard skin and are seeded [14]. Their consumption in the form of raisins could be more accepted by consumers, and these grape cultivars may be grown with more added value in humid ecologies if their potential as raisins was shown to be high and technologically feasible. The grape cultivars used in this study were selected among cultivars that are resistant/tolerant to fungal diseases. Apart from the berry colour and cultivar type, the total phenolic content of grapes changes depending on many

internal and external factors and cultural practices. This situation directs physiologists and breeders to obtain new lines with increased antioxidant capacities by controlling these factors [15]. Due to the positive effects of phenolic compounds and antioxidants on human health, the demand for products rich in phenolics has increased [16], and phenolic components are considered important quality parameters for plant foods [17]. Therefore, it is important to determine their amounts in fresh grapes and raisins. Certain sensory and quality characteristics of grape berries from nine of the most suitable cultivars, which can be grown in especially humid regions with very light pesticide application, were compared. Moreover, the sensory and quality characteristics of raisins obtained from these grape cultivars (belonging to different *Vitis* species) were determined by using an energy-saving heat pump dryer specifically designed for this study. The number of grape species and cultivars grown in humid regions is very limited due to fungal diseases. In addition, the fresh consumption of grapes from cultivars grown in these regions is limited due to their short shelf life and thicker skins. In this study, we aimed to determine the raisin production potential of different cultivars which might be grown in humid regions, and some physical, chemical and sensorial properties of raisins from these cultivars were reported.

## 2. Materials and Methods

### 2.1. Grape Cultivars

Grapes were obtained from the experimental vineyard of the Yalova Atatürk Horticultural Central Research Institute (Yalova, Turkey), northwest Turkey. Yalova Province is located between 28°45′ and 29°35′ East longitudes, 40°28′ and 40°45′ North latitudes; the average height of the vineyard area from the sea is 2 m. The climate in Yalova is cold and rainy in the winter and hot in the summer, but the relative humidity is usually higher than the climate in the south of the country. While the annual average precipitation is 750–800 mm, the relative humidity is in the range of 75–80%. This situation creates a suitable environment for fungal diseases in the summer months when vegetative growth is most active and the fruits ripen. Grape cultivars were grown in the Yalova province, northwest Turkey. Vines were approximately 6 years old and grown under identical conditions. They were planted with a distance of 3 m between rows and 2 m above the rows with a pergola training system. Grapes belonging to the production season of 2021 were analysed in this study. The major characteristics of the grape cultivars used in the study are given in Table 1 and bunches of grapes from these cultivars are visually shown in Figure 1. Also, pictures of raisins obtained from the different grape cultivars are given in Figure 2.

**Table 1.** Major characteristics and origins of grape cultivars used in this study.

| Cultivar | Species | Origin of Material | Berry Colour | Special Flavour | Seed Status |
|---|---|---|---|---|---|
| Mars Seedless | | PGRU * | Black | Foxy | Seedless |
| Campbell Early | | GBARES ** | Black | Foxy | Seeded |
| Muscat Bleu | Interspecies | IGBG *** | Black | No | Seeded |
| Glenora | | PGRU | Rose | No | Seedless |
| Kay Gray | | PGRU | Yellow/Green | Foxy | Seeded |
| Philipp | | IGBG | Black | No | Seeded |
| Kishmish Vatkana | | IGBG | Black | No | Seedless |
| Rhea | *V. vinifera* | IGBG | Rose | No | Seedless |
| Özer Beyazı | | VRI **** | Yellow/Green | No | Seedless |

* PGRU, USDA Plant Genetic Resources Unit Geneva, NY, USA; ** GBARES, Gyeongsangbuk-do Agricultural Research and Extension Services, Daegu, Korea; *** IGBG, Institute for Grapevine Breeding, Geilweilerhof, Germany; **** VRI, Viticulture Research Institute, Tekirdağ, Turkey.

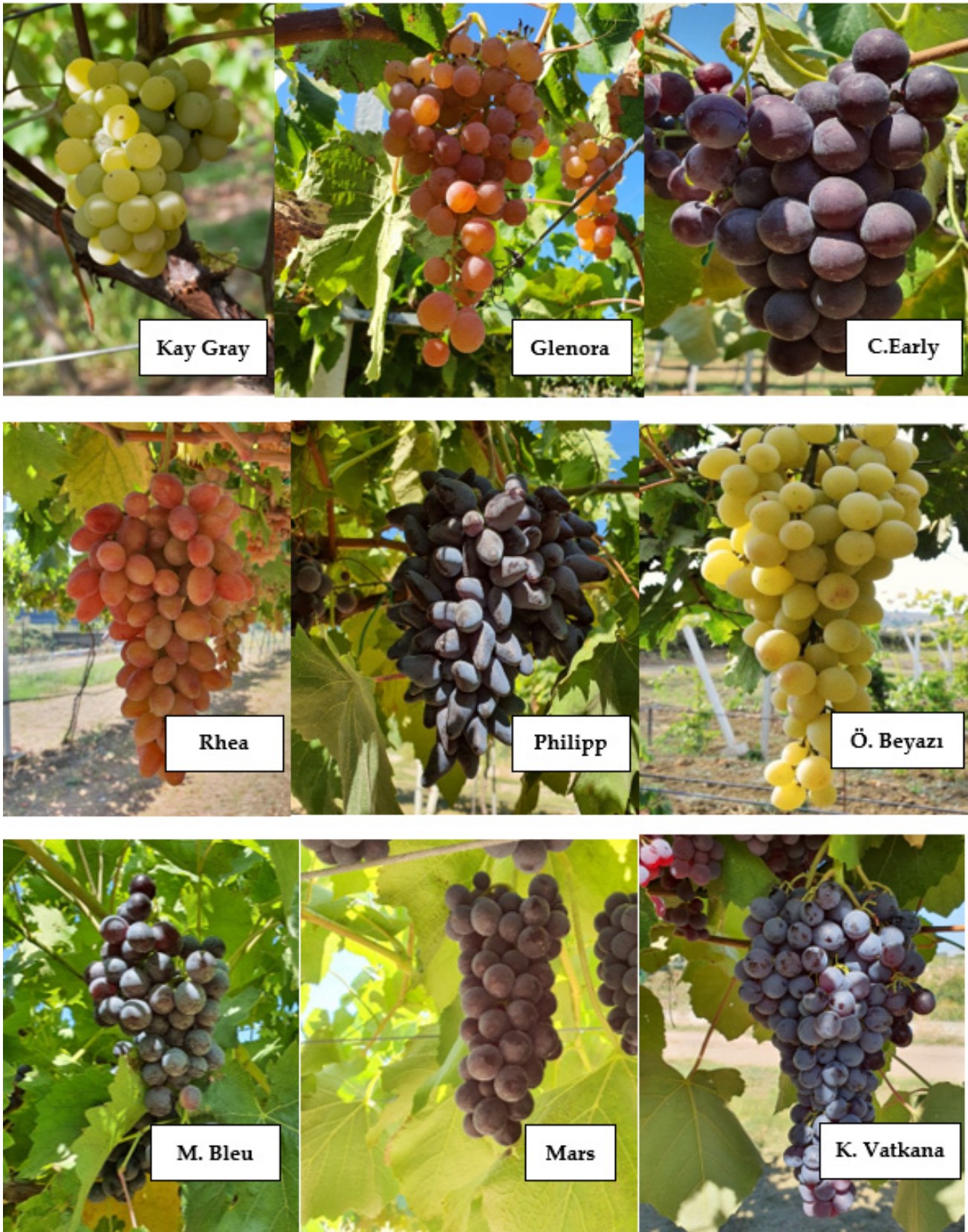

**Figure 1.** Bunches of grapes from the nine different grape cultivars used in this study.

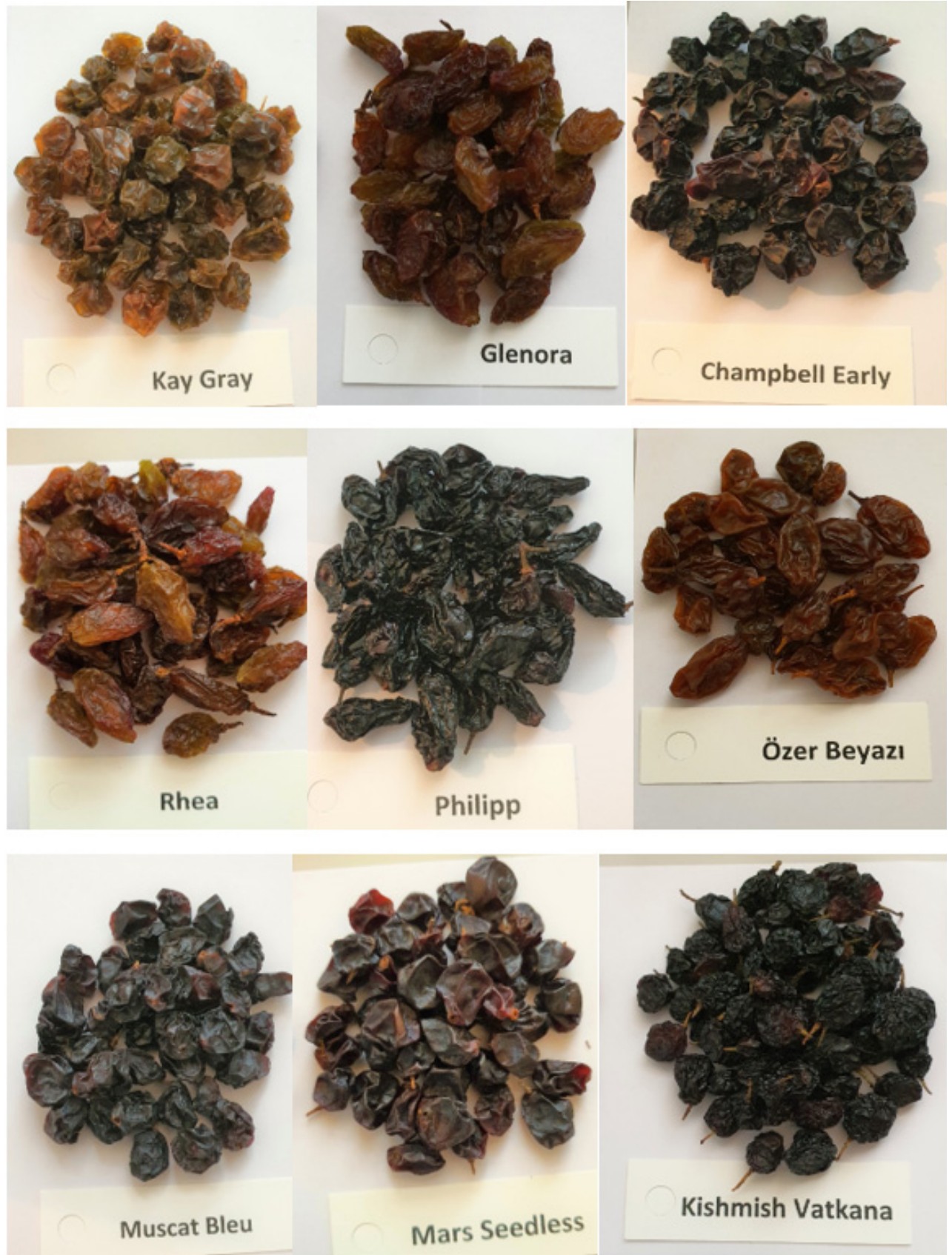

**Figure 2.** Pictures of raisins obtained from the nine grape cultivars by drying them in an air-pump dryer.

## 2.2. Drying Process

As a pre-treatment, fresh grape berries were immersed in potash solution (5% $K_2CO_3$ + 1% olive oil) for a short time (4–6 s) to remove the outer waxy layer from berries before drying. With this application, the monomolecular structure of the wax layer on the berry surface was deteriorated and the water permeability of the berry skin increased [18]. Grape berries were then strained and placed in drying boxes, and later they were taken to a closed-circuit heat pump drying (HPD) system which was specially designed for this study. This closed-circuit HDP system in particular was selected because it saves energy and can be easily controlled. The closed circuit HPD system consisted of five main parts: a compressor, condenser, evaporator, expansion valve and drying cabinet (Figure 3).

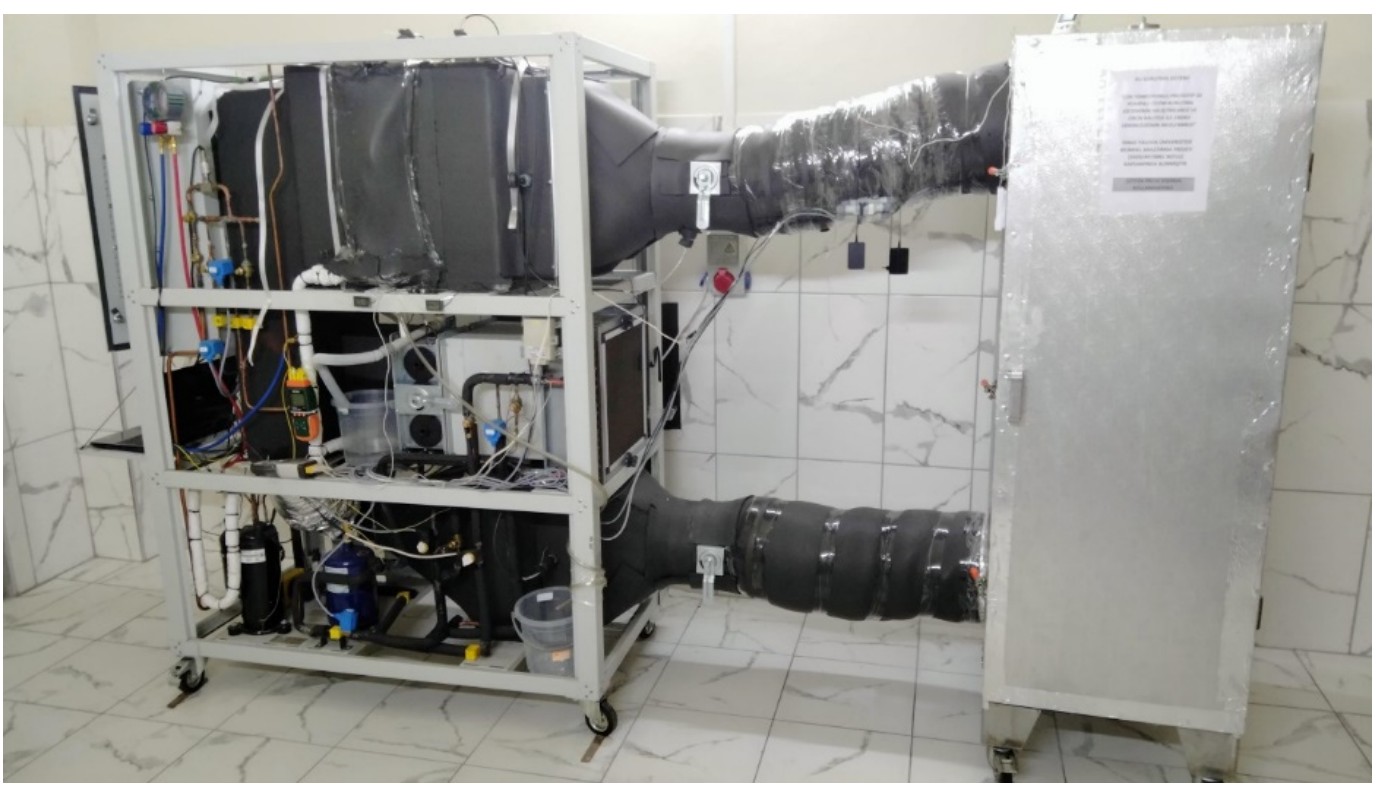

**Figure 3.** The closed circuit HPD system specially designed for this study.

The drying of berries was carried out in an air-pump dryer at 45 °C with an air velocity rate of 3 m/s until a similar water content was achieved for all grapes (about 16% moisture or 0.19 g/g).

The moisture content of each sample was achieved. During drying, samples were regularly weighted by a digital balance to check the target water content. The properties of fresh grapes were compared with those of dried grapes. The moisture ratio (MR) was calculated as the ratio between the actual (Mt) and the initial (M0) moisture content. Results were reported as the average of three sets of experiments.

## 2.3. Moisture Content, Water Activity and Brix Values

For these analyses, the optimum harvest time of grape berries was checked by sampling regularly every week, starting from the 3rd week after veraison. Ten bunches of each grape cultivar were selected from different vines to represent the cultivar and brought to the laboratory. Berries were separated from their bunches and cleaned. 30 berries were then randomly selected, and these berries were divided into three groups of ten and analysed in three replications for fresh berries. Likewise, after drying, 30 raisins were randomly

selected, and these berries were divided into three groups of ten and analysed in three replications. The results are given as the mean of these replications.

All experiments were carried out in triplicates, and results were means of two parallel measurements. The water activity values of samples were determined by a water activity measuring device (LabMASTER, Novasina, Lachen, Switzerland). Samples were placed in a special container supplied by the manufacturer of the device, and this was then placed in a sealed steel cell. The water activity value of a sample, which came into equilibrium with the environment at room temperature, was read from the screen of the device and recorded. Raisins were crushed into pulp by using a mortar, and two grams of this pulp was spread out in a tarred stainless capsule and then dried in a drying oven at 80 °C until a constant weight was reached. The moisture content (%) was calculated gravimetrically by using the Equation (1).

$$\text{Moisture Content (\%)} = ((Wbd - Wad)/Wbd)) \times 100 \tag{1}$$

where Wbd is the weight of sample before drying and Wad the weight of sample after drying.

A digital refractometer (PAL-BX/ACID2, Atago Co. Ltd., Tokyo, Japan) was used to determine the soluble solid content of the samples (i.e., Brix values (%)) in fresh grapes. The brix values of raisins were calculated by using the Equation (2).

$$\text{\% Soluble Solid Content (g/100 mL)} = (B \times V)/S \tag{2}$$

where B is the degree of Brix determined in a diluted sample, V is the volume in which the sample was diluted (mL) and S is sample amount (g).

### 2.4. Extraction of Bioactive Compounds

Extraction of bioactive compounds was obtained according to Thaipong et al. [19]. Samples (3 g) were partially ground by a laboratory blender and homogenized with 25 mL of methanol (chromatographic grade) in a Silverson brand homogenizer for 2 min. The same method used for the extraction of fresh grapes was used for the extraction of raisins. The homogenized sample was kept at +4 °C for a night. The next day, it was centrifuged at 15,000× *g* for 20 min in an Eppendorf centrifuge. Supernatant was collected in amber bottles with a glass Pasteur pipette and stored at −20 °C until analysis. This extract was used in the analyses of total phenolic contents and antioxidant activity assays, which were performed using three different methods (2,2-diphenyl-1-picrylhydrazyl (DPPH), ferric reducing antioxidant potential assay (FRAP) and Cupric reducing antioxidant capacity assay (CUPRAC)) to determine the bioactive properties of fresh berries and their raisins.

### 2.5. Total Phenolic Content

The total phenolic contents of samples were determined by the Folin-Ciocalteu (FC) method [20]. Briefly, 30 µL extract and 150 µL FC reagent were consecutively transferred into tubes including 2.37 mL deionized water. After 8 min, 450 µL saturated $Na_2CO_3$ was added to the mixture. The same procedure was run to prepare blank using 30 µL deionized water instead of extract. Absorbance values were read at 750 nm using a spectrophotometer against the blank after 30 min incubation at 40 °C. Various concentrations of gallic acid solutions (50, 100, 200, 300, 400 and 500 mg/L) were used to plot a calibration curve. Results were expressed as mg gallic acid equivalent (GAE) per 100 g dry weight (db).

### 2.6. Antioxidant Activity Assays

The DPPH (2,2-diphenyl-1-picrylhydrazyl) assay procedure described by Thaipong et al. [19] was used with slight modifications. DPPH stock solution was prepared as 24 mg/100 mL methanol (chromatographic grade) and stored at −24 °C before use. Working solution was prepared by diluting the stock solution with methanol to the final absorbance of 1.20 ± 0.02 at 515 nm. For the calibration curve, Trolox® solution with a final concentration of less than 50 µM in a spectrophotometer cuvette was used. In the experiments, extract or standard (150 µL) and DPPH working solution (2850 µL) were

mixed in a test tube, and the reaction was continued for 60 min in a dark environment. At the end of this period, the absorbance of coloured solutions was read at 515 nm, and results were expressed as mmol TE per 100 g. Extracts were diluted if their absorbance reading was over the linear range of the calibration curve.

The ferric reducing ability of plasma (FRAP) assay procedure described by Thaipong et al. [19] and Katalinic et al. [21] was used to determine the antioxidant activities of fresh berries and their raisins. Stock solution included 300 mM acetate buffer (3.1 g of $C_2H_3NaO_2 \cdot 3H_2O$ and 16 mL of $C_2H_4O_2$), pH 3.6 and 10 mM TPTZ (2,4,6-tripyridyl-s-triazine) solution in 40 mM HCl and 20 mM $FeCl_3 \cdot 6H_2O$ solution. Fresh working solution was prepared by mixing 25 mL of acetate buffer, 2.5 mL of TPTZ solution and 2.5 mL of $FeCl_3 \cdot 6H_2O$ solution and then kept at 37 °C before use. This reagent was added to sample extracts of 150 μL, and the reduced form of blue colour was read at 593 nm after 30 min. Trolox® was used as a standard, and the ferric reducing power of extracts was calculated by its calibration curve. The standard curve was linear between 25 and 800 μM Trolox®. Results are expressed in μmol TE per 100 g. Extracts were diluted if their absorbance reading was over the linear range of the calibration curve.

Cupric reducing antioxidant capacity (CUPRAC) assay was used to determine the antioxidant capacity of fresh berries and their raisins. This method is used to measure the ability of polyphenols to reduce copper (II) or cupric ion, vitamin C and vitamin E [22], and copper (II)-neocuproine (Cu (II)-NC) reagent I is used as a chromogenic oxidizing agent. In this method, an antioxidant solution is mixed with a copper (II) chloride solution, a neocuproine alcoholic solution and an ammonium aqueous buffer at pH 7, and subsequently the absorbances of mixtures are determined at 450 nm after 30 min. The antioxidant activities of grape and raisin samples by the CUPRAC assay were determined as follows: 1 mL each of copper (II) chloride solution (10 mM), neocuproine solution (Nc) of 7.5 mM and ammonium acetate ($NH_4$ Ac) buffer (pH 7) solutions were mixed in a test tube, and extract (or standard) solution (x mL) and $H_2O$ ((1.1 − x) mL) were added to the test tube to make the final volume 4.1 mL. Test tubes were screw-capped and left for an hour at room temperature, and then the absorbance at 450 nm was recorded against a reagent blank. The standard calibration curve of each antioxidant compound was constructed with Trolox® in this manner as absorbance versus concentration. The molar absorptivity of the CUPRAC method for each antioxidant was found from the slope of the calibration line concerned, and the antioxidant activity values of fresh grape berries or their raisins were expressed as mg Trolox equivalents per 100 g.

### 2.7. Colour Analyses

For all samples, colour was assessed by the CIELab space with a chromameter (CR–400, Minolta Co., Tokyo, Japan) [23]. The parameters that define the CIELab space are: rectangular coordinates such as red/green colour component (a*), yellow/blue colour component (b*) and lightness (L*) and the cylindrical coordinates of chroma and hue angle. The colour differences between two different products were calculated in CIELab units, using the parameter (ΔE*), calculated as $\Delta E^* = (\Delta L^{*2} + \Delta a^{*2} + \Delta b^{*2})^{0.5}$.

### 2.8. Sensory Evaluation

Experienced panellists (five male and five female members of staff of the AHCRI (Yalova Atatürk Horticultural Central Research Institute, Yalova, Turkey) in the range of 30–50 years old) participated in sensory evaluation sessions. Prior to testing, panellists received a semi-training session related to the seven sensory attributes of raisins such as colour, appearance (such as colour homogeneity, raisin size, raisin size homogeneity and size of wrinkles), odour, taste (such as sweet, sour, bitter and astringent), hardness, chewiness (textural properties) and overall likability, so that each panellist could thoroughly discuss and clarify the attributes of samples to be evaluated [24]. All trials were assessed in consistent conditions, including a controlled temperature and adequate illumination, free from odour. Drinking water and crackers were provided between samples to cleanse the

palate. After evaluating each sample, panellists were asked to assign a numerical value between 1 (dislike very much) and 10 (like very much) for raisins.

### 2.9. Statistical Analysis

Statistical analyses were performed by using the SAS statistical software (The SAS System for Windows 9.0, Chicago, IL, USA). Results were expressed as mean ± standard deviation, and the Duncan multiple comparison test for $p < 0.05$ was used to analyse significant differences of variance (ANOVA).

### 3. Results

The drying of berries was carried out in an air-pump dryer at 45 °C with an air velocity rate of 3 m/s until a similar water content for all grapes (about 16% moisture or 0.19 g/g moisture content of sample) was achieved (Table 2). During drying, the weight of samples was regularly monitored by a digital balance to check if the target water content was reached. Pictures of raisins obtained from different grape cultivars are shown in Figure 2.

**Table 2.** Initial weight of fresh berries and final moisture values of their raisins after drying in an air-pump dryer.

| Cultivars | Total Drying Time (h) * | Initial Weight (kg) | Moisture Content (%) | Moisture Ratio | Moisture Content (g/g) |
|---|---|---|---|---|---|
| Muscat Bleu | 61.67 | 1.40 | 15.97 | 0.046 | 0.190 |
| Mars Seedless | 45.67 | 1.50 | 15.98 | 0.051 | 0.190 |
| Kay Gray | 93.50 | 1.65 | 15.99 | 0.057 | 0.190 |
| Glenora | 85.17 | 1.67 | 15.90 | 0.058 | 0.189 |
| Rhea | 42.50 | 1.56 | 15.95 | 0.053 | 0.190 |
| Campbell Early | 69.67 | 1.35 | 15.93 | 0.044 | 0.189 |
| Kishmish Vatkana | 37.67 | 1.64 | 15.98 | 0.057 | 0.190 |
| Philipp | 57.67 | 1.42 | 15.98 | 0.047 | 0.190 |
| Özer Beyazi | 79.00 | 1.33 | 16.01 | 0.044 | 0.191 |

* Drying time represents the time to reach a moisture content of about 16% for each cultivar at 45 °C with an air velocity rate of 3 m/s.

The bioactive contents of fresh berries from different grapes cultivars values are presented in Table A1. The total phenolic contents of fresh berries ranged from 7.20 (Rhea) to 12.89 g GAE per 100 g (Philipp). In terms of dry weight, the total phenolic contents of Muscat Bleu (65.96 g GAE/100 g), Philipp (64.88 g GAE/100 g) and Campbell Early (64.53 g GAE/100 g) were the highest ($p < 0.05$). The fresh berries of the Kishmish Vatkana cultivar had the lowest antioxidant activity according to the DPPH assay (525.81 mmol TE/100 g) ($p < 0.05$) while the berries of other cultivars had a similar antioxidant activity ($p > 0.05$). The berries of Özer Beyazı and Campbell Early had an antioxidant activity value determined by DPPH assay of about 3550 mmol TE/100 g on a dry basis, and were found to have similar antioxidant activity values to the Muscat Bleu and Philipp cultivars ($p > 0.05$). In terms of the antioxidant activity values determined by FRAP assay, the fresh berries of Philipp (3120.00 mmol TE) and Muscat Bleu (2922.78 mmol TE/100 g) had the highest antioxidant activity ($p < 0.05$). On a dry basis, the Philipp and Muscat Bleu berries had an antioxidant activity about 15,700 and 15,100 mmol TE/100 g db, respectively, which were found to be similar ($p > 0.05$). The berries of the Glenora, Kay Gray, Rhea and Kishmish Vatkana cultivars had a similar antioxidant activity, as determined by FRAP assay, on a dry basis ($p > 0.05$). In terms of antioxidant activity by CUPRAC assay, the Muscat Bleu, 'Philipp' and Campbell Early cultivars had a much higher value than the rest of the cultivars in terms of both fresh and dry weight. The interspecies hybrid 'Philipp' cultivar stood out among the other cultivars with its 446.49 mg Trolox/100 g fresh weight and its 2246.54 mg Trolox/100 g db (Table A1). According to the CUPRAC assay, the berries of Philipp and

Muscat Bleu cultivars had a similar antioxidant activity value in terms of both fresh and dry weight ($p > 0.05$).

The bioactive contents of raisins from different grape cultivars were expressed based on both fresh and dry matter. Values based on dry matter are mostly useful to compare the bioactive contents of raisins such as their total phenolic contents and antioxidant activities determined by DPPH, FRAP and CUPRAC assays, while those based on fresh weight are important for a nutritional point of view. When the values of bioactive contents of raisins from different grape cultivars were compared, trends similar to fresh grapes were obtained. The bioactive contents of raisins from different grape cultivars based on fresh and dry weights are shown in Table A2. The total phenolic contents of raisins based on fresh weight ranged from 12.76 to 13.58 g GAE/100 g, while the highest value on a dry weight basis was 19.30 g GAE/100 g for the raisins of the Özer Beyazı cultivar ($p < 0.05$). The results of the DPPH assay showed that the antioxidant activity values of raisins based on fresh weight changed in a relatively narrow range, while the highest value based on dry weight was found for the raisins of Özer Beyazı (991.01 mmol TE/100 g). The antioxidant activity results of the FRAP assay based on fresh weight ranged from 1981.11 to 6962.50 mmol TE/100 g. The raisins of the Philipp cultivar had an antioxidant activity of 7893.51 mmol TE/100 g db, which was found to be significantly higher than that of other cultivars, with an exception in the raisins of Muscat Bleu ($p < 0.05$). The range for the antioxidant activity results of the CUPRAC assay in terms of both fresh and dry weight was found to be much higher than the ranges for the other assays (Table A2). The highest antioxidant activity determined by the CUPRAC assay was found for the raisins of the Philipp cultivar ($p < 0.05$). The interspecies hybrid Philipp cultivar stood out among the other cultivars, with values of 4505.21 mg Trolox/100 g fresh weight and 5109.87 mg Trolox/100 g db.

The moisture contents, water activity and soluble solid contents of fresh grapes and their raisins from nine different cultivars are shown in Table A3. The moisture contents of fresh grapes from different cultivars ranged from 76.66 (Glenora) to 81.49% (Campbell Early), while those of raisins ranged from 11.80 (Philipp) to 33.54% (Özer Beyazı). The water activity of fresh grapes was around 0.85, while the water activity of raisins ranged from 0.71 (Özer Beyazı) to 0.42 (Kay Gray). In general, all raisins had a water activity value below 0.71, which could be considered as safe for storage at room temperature. Regarding the water-soluble solids content (°Brix), the Kishmish Vatkana cultivar had the highest value as both fresh grapes (21.70%) and raisins (65.10%) ($p < 0.05$) (Table A3).

The colour properties of fresh grapes and their raisins from nine different grape cultivars are presented in Table A4. The fresh grapes of the Kay Gray and Glenora cultivars had the highest lightness (L*) values, at 55.51 and 53.50, respectively ($p < 0.05$). On the other hand, the raisins of Glenora, Mars Seedless, Muscat Bleu had a CIE L* colour value (about 51.50) which was higher than those of the other cultivars ($p < 0.05$). The highest value for a red colour component (positive CIE a* values) was obtained for the fresh grapes of the Rhea cultivar (3.60), while the highest a* values were found for the raisins of the Özer Beyazı (2.67) and Rhea (2.41) cultivars ($p < 0.05$). The highest yellow colour component (positive CIE b* values) was obtained for the fresh grapes of Özer Beyazı (4.42) ($p < 0.05$), while among the raisins, the highest values were found for Kay Gray (4.60), Rhea (4.47) and Özer Beyazı (3.39) ($p < 0.05$). In terms of cylindrical coordinates such chroma values, the fresh grapes of Özer Beyazı had a chroma value of 4.74, which was significantly higher than those of the other cultivars, with the exception of Rhea (4.09) ($p < 0.05$). The chroma values of raisins ranged from 0.31 (Philipp) to 5.08 (Rhea). Based on the hue angle values, the highest value was obtained for the raisins of the Kay Gray cultivar (1.36), with an exception being those of the Rhea cultivar ($p < 0.05$). The colour difference value (ΔE*) indicates the effect of drying conditions on the overall colour properties of a dried product, and the colour properties of the fresh grapes were used as initial values. According to the results of the colour difference values calculated in CIELab units, the cultivars with the highest colour difference values were Kay Gray (17.22) and Muscat Bleu (15.91) ($p < 0.05$), and their colour difference values were significantly higher than those of the cultivars with the least

colour changes, such as Glenora (2.70), Campbell Early (3.96), Rhea (4.62) and Kishmish Vatkana (5.66) ($p < 0.05$) (Table A4).

The sensory evaluation included the raisins of nine different grape cultivars according to seven different aspects including their colour, appearance, odour, taste, hardness, chewiness and overall likability. The sensory colour scores of raisins from the Kay Gray and Mars Seedless cultivars were 6.50, which was significantly lower than those of the Kishmish Vatkana (9.00), Rhea (8.80), Philipp (8.20), Campbell Early (8.00), Glenora (7.70) and Özer Beyazı (7.70) cultivars ($p < 0.05$). Regarding appearance, the raisins of the Kishmish Vatkana, Rhea, and Glenora cultivars received the sensory scores of 9.00, 8.80 and 8.30, respectively, which were significantly higher than those of the Kay Gray, Mars Seedless, Philipp and Campbell Early cultivars ($p < 0.05$). In terms of the odour scores, the raisins of the Rhea cultivar (8.80) received the highest score, with the exception of Kishmish Vatkana (8.50) ($p < 0.05$). The raisins of the Rhea (9.60), Kishmish Vatkana (9.30) and Glenora cultivars (8.90) received the highest taste scores from the panellists ($p < 0.05$). In a similar way, the raisins of the Kishmish Vatkana (9.10), Rhea (9.10) and Glenora cultivars (8.90) received the highest hardness scores ($p < 0.05$). The chewiness scores of the raisins of these three cultivars (>9) were also the highest ($p < 0.05$). Finally, the panellists' overall likability scores for the raisins of nine different grape cultivars were ≥6.20, and the raisins of the Rhea (9.50) and Kishmish Vatkana cultivars (9.40) were found to be the most-liked raisins ($p < 0.05$). In comparison to these two seedless cultivars, the raisins of the two seeded cultivars, Philipp and Kay Gray, were perceived as the least liked, with an overall likability score ≤6.30 (Table A5).

## 4. Discussion

In this study, the phytochemical contents of grape berries from nine different grape cultivars and their raisins, which were dried in an air-pump dryer to a moisture content of about 16%, were mainly determined. The results showed that the berries of red/black cultivars generally had a higher antioxidant activity than those of yellow/green cultivars, with the exception of the Özer Beyazı cultivar (Table A1). Numerous studies are available on the bioactive contents of native and hybrid grape cultivars, and most of the studies are focused on the phenolic contents of the different parts of grape berries, such as their pulp, skin and seeds. In our study, the major aim was to evaluate the bioactive contents of raisins from different grape cultivars with a high raisin production potential; therefore, we determined the bioactive contents of whole grape berries before the production of raisins using an air-pump dryer at 45 °C. According to the FRAP and CUPRAC assays, the berries of the Muscat Bleu, Philipp and Campbell Early cultivars with red/black berries had a comparatively higher antioxidant activity value than those of the others cultivars based on fresh and dry weight. The ranges for the TPCs of whole grape berries from nine different cultivars were from 7.20 to 12.89 g GAE/100 g in terms of fw and from 33.27 to 65.96 g GAE/100 g in terms of dry weight. The antioxidant activity values of fresh berries as determined by the DPPH and FRAP assays ranged from 2334.24 to 3556.59 and from 4892.02 to 15,699.63 mmol TE/100 g db, and the results of the FRAP assay indicated that the antioxidant activity values of the berries of Muscat Bleu, Philipp and Campbell Early were exceptionally higher than those of the others. Most of the time, the phytochemical contents of seed, skin and pulp of different grape cultivars are separately reported in the literature, which makes the comparison of their values with whole berries difficult. Moreover, irregularities in the reporting of the results relating to antioxidant activity assays such as DPPH, ABTS and FRAP is widespread in the literature, which might greatly complicate the comparison of different data. TPC assay results have consistently been reported as mg GAE per weight, which makes any comparison highly possible. In a study by Xu et al. [25], the TPCs of grape seeds from 18 Oriental and North American *Vitis* Species/hybrids, and *V. vinifera* grape cultivars changed significantly from 1599 to 9928 mg GAE/100 g db, and the seeds of Cabernet Sauvignon had the highest, followed by Muscadines, while the EuroAsian or Euro-American hybrids were

in between their parents. They reported that the seeds of Cabernet Sauvignon had the highest antioxidant value (422, 650 and 605 μM TE/g db according to the DPPH, ABTS and FRAP assay, respectively, which was followed by those of Muscadines, while oriental *Vitis* grapes had the lowest. Farhadi et al. [26] reported that the skins of grape berries from the Ghara Shani (black) cultivar grown in West Azerbaijan (Iran) had the highest TPC (1205 ± 141 mg GAE/g db), showing a DPPH radical scavenging activity of up to 95%. Among 30 grape cultivars of white, red and pink grapes (28 interspecies hybrids and 2 *V. vinifera* L.) grown in Poland, Samoticha et al. [27] reported that grapes had a TPC ranging from 1037 (Cascade) to 5759 mg GAE/100 g db (Regent). Red grape cultivars of Roesler, Rothay and Swenson Red generally had the highest content of bioactive compounds, and their antioxidant activities determined by ABTS, FRAP and ORAC assays were 24, 12 and 53 mmol TE/100 g db, respectively, which were noticeably higher than those of the other cultivar. Studying the variety differences of fresh grape phenolic profiles in eleven grape cultivars, Li et al. [28] reported that Muscat Kyoho extracts had the highest total phenolic content in skins (10.53 mg GAE/g fw) and pulps (1.13 mg GAE/g fw), and the highest DPPH radical scavenging capacity in the free phenolics of their skins (EC50 = 11.7 μg/mL), with the lowest related value (EC50 = 19.2 μg/mL) being for Shine Muscat. They reported the DPPH radical scavenging ability of free phenolics to be about 1.8 times stronger than that of bound phenolics in fresh grape skins. Fidan et al. [29] studied the effect of grape cultivar, molecular concepts and agro-climatic factors on the bioactive constituents of grape seeds of four different grape cultivars from five different locations in Siirt (Turkey) and found that the TPCs of Rutik and Gadüv grapes were 70.9 and 62.9 mg GAE/g, respectively. The inhibition rates for DPPH radicals for these two cultivars were about 95%, and their antioxidant activity values determined by FRAP assay were 69.0 and 67.6 μmol $Fe^{+2}$/g, respectively. Determining the antioxidant compounds and antioxidant activity of six red-skinned table grapes grown using rain-shelter cultivation in the Chengdu Plain, south China, Shen et al. [30] reported that the ranges of the TPCs were 280–578 in skins, 76–140 in pulps and 2752–4062 mg GAE/kg fw in seeds, and the antioxidant activity of the Red Globe, Jintianmeizhi and Hongbaladuo cultivars in the DPPH assay were higher than those of others. The ranges for the antioxidant activity in the FRAP assay were reported as 914–1228 in skins, 190–270 in pulps and 2805–4083 μmol TE/g fw in seeds. Doshi et al. [31] determined the phenolic composition and antioxidant potential of different parts of a commercially popular grape variety, Kishmish Chornyi (Sharad Seedless), at various stages of maturation and reported that berries at the initial developmental stage showed significantly high contents of total phenolics (95 mg GAE/g). Together with berry stems, they had high contents of total phenolics, flavonoids, flavonols, flavan-3-ols and antioxidant activity in the FRAP assay. However, with maturation, they had a significant gradual reduction in these polyphenol contents.

The phytochemical or bioactive contents of grape berries can be influenced by a number of factors that can be genetic, related to cultivation or climatic. In this sense, continuous efforts to produce grapes with a superior raisin production potential are highly noticeable in the literature as well as in industry. In this study, the TPCs of fresh berries was the highest (12.89 g GAE/100 g fw) for the Philipp cultivar but the highest value (19.30 g GAE/100 g db) was found for the raisins of the Özer Beyazı cultivar, which exhibited the highest antioxidant activity in the DPPH assay (991.01 mmol TE/100 g db). Studying the antioxidant activity (by ABTS assay) and total and individual phenolic compounds of six raisin grape cultivars (commercial) and ten new raisin grape selections, Breksa et al. [32] reported a 3.6-fold difference between the lowest and highest concentrations among white grape cultivars and concluded that genetic factors (other than skin colour) and stress levels could be significant in terms of the phenolic content of a cultivar. They found that the antioxidant activity of all samples ranged from 7.7 to 60.9 μmol Trolox/g db, while a TPC ranged from 3.16 to 11.41 g GAE/100 g db. Thompson Seedless grapes had the lowest TPC, 357.7 mg GAE/100 g db, among commercial cultivars. The TPCs of the grape berries used in our study ranged from 33.27 to 65.96 g GAE/100 g db, and they were six to ten-fold

higher than those reported by Breksa et al. [32], which could be explained by differences in the cultivars, cultivation method, climate or seed-status of the cultivar besides the phenolic extraction conditions during analysis. In a study by Liu et al. [33], the total FRAP values of lipophilic, hydrophilic and insoluble-bound fractions from 30 grape cultivars ranged from 1.29 to 11.77 µmol $Fe^{2+}$/g fw, while the range was 0.34–4.84 µmol Trolox/g fw for the TE antioxidant capacity assay and 0.29–1.41 mg GAE/g fw for the total phenolic content. The variability in TPC values was attributed to the genetic and environmental factors of growing location, such as the climate, soil composition, temperature, ripening stage and varietal differences of grape cultivars. Xia et al. [34] reported that the total phenolic contents ranged from 12.28 to 95.23 in the seeds and from 7.94 to 44.64 mg GAE/g db in the skins of 31 different grape cultivars. European cultivars such as Cabernet Sauvignon, Muscat Hamburg and Merlot had the highest values of phenolic contents and antioxidant properties in seeds, followed by Muscadine, American and Oriental grapes. On the other hand, insignificant differences in these values were reported for the skin extracts of different grape cultivars. The phenolic composition of grapes can be significantly influenced by any variation in climatic conditions, even for the same cultivar [35] and in genotypes of a cultivar [36]. Studying the phytochemical properties of three white-skinned grapes and their coloured genotypes at different growth stages, Niu et al. [36] reported that coloured genotypes had a significantly higher content of malvidin 3-O-glucoside than white-skinned grapes, while the difference in their delphinidin 3-O-glucoside contents was insignificant. With the exception of cultivars such as Pinot Blanc and Pinot Noir, the antioxidant activity of white-skinned grapes determined by the DPPH and CUPRAC assays was found to be similar to their coloured genotypes at any growth stage, including in terms of their pulp and seeds. They also noted that the antioxidant capacity of grape cultivars was not always in correlation with the skin colour of grape berries. Similar to that, we also found that the antioxidant activity values for the Kay Gray and Özer Beyazı cultivars with yellow/green berries were compatible with those with black or rose berries (Table A1). Callaghan et al. [37] compared the total antioxidant capacities (TAC) of purple, red, green (seedless) and concord (with seeds) table grapes by using the CUPRAC assay and found that the total antioxidant activities of Concord and purple grapes were significantly higher than those of red or green grapes. Concord grapes have fairly large, semi-hard seeds, while others have several small, soft seeds. They reported that antioxidants are primarily located in the skins of the purple and red grapes, while in the Concord and green grapes, they are equally divided between the skin and pulp.

The bioactive contents of raisins (Table A2) are expressed in terms of both fresh and dry weight in this study since the former could be important for a nutritional point of view, while the latter is usually critical for comparing the contents of different grape cultivars. Since the dry matter content of raisins ranged from 11.80 to 33.54% (Table A3), their bioactive contents based on fresh weight were somewhat close to those based on dry weight for most of cultivars. The total phenolic content of raisins ranged from 15.0 to 19.3 g GAE/100 g db, while the ranges of their antioxidant activities, as determined by the DPPH, FRAP and CUPRAC assays, were 723.6–991.0 mmol TE/100 g, 2419.8–6290.8 mmol TE/100 g and 115.7–5109.9 mg Trolox/100 g db. The antioxidant activity of raisins determined by the DPPH and Folin-Ciocalteu [38] and ORAC [39] assays has been reported previously. Çelik [7] determined the effect of three local cultivars of *V. vinifera* (Osmanca, Razaki and Gelin) and pre-treatment solutions on the drying period and raisin grape quality and reported the highest total phenolic content in the raisins of Osmanca (1107.3 mg GAE/100 g), while the TPCs of the raisins from Razaki and Gelin were 681.6 and 670.0 mg GAE/100 g, respectively. In our study, the raisins from different grape cultivars contained TPCs higher than those reported by Celik, which might result from the differences in the grape cultivars, cultivation, climate, seed-statues of cultivars, drying conditions and phenolic extraction conditions. Kelebek et al. [40] reported that the antioxidant activity of red raisins was better than white ones. In our study, the raisins of red/black grape cultivars such as Muscat Blue, Philipp, Campbell Early and Mars Seedless had higher TPCs than

those of yellow/green cultivars such as Özer Beyazı and Kay Gray in terms of fresh or dry weight ($p < 0.05$). On the other hand, this trend was not true for all cultivars since the raisins of Kishmish Vatkana, which is a black/red cultivar, had a lower TPC than those of a white Kay Gray cultivar on a fresh weight basis ($p < 0.05$). Differences in grape cultivars such as pulp colour and density, seed and flavour status are mostly likely to be the reason for variances in their TPCs.

Several studies are available on the TPCs and antioxidant activities of raisins from different grape cultivars and the effect of drying and pre-treatment conditions on these parameters in the literature. Meng et al. [41] determined the bioactive contents of the raisins of nine grape genotypes from Xinjiang (China) by using several assays including FC, DPPH, CUPRAC, Potassium FRAP, and hydroxyl radical scavenging capacity (HRSC). The Desert King cultivar had the highest content of total phenolics (678.4 mg GAE/100 g db). The highest TPC was 678.4 mg GAE/100 g db in the raisins of Desert King, which also showed the highest antioxidant capacities in the DPPH, CUPRAC and Potassium FRAP assays: 1118.9 µg Trolox/g db, 1298.3 µg Trolox/g db and 0.960 (absorbance), respectively. On the other hand, the highest HRSC assay result was found for the raisins of Red man-aizi. Mnari et al. [42] determined the bioactive contents of raisins from four (*V. vinifera* L.) Tunisian cultivars (Chriha, Razeki, Assli and Meski). Their total phenolic contents ranged from 401.5 to 534.2 mg GAE/g db, and Chriha had the highest, followed by Assli, Meski and Razeki. Meski exhibited the highest activity, as determined by the DPPH assay. The extraction solvents and methods used in a study may have influenced the total phenolic contents of raisins [43] in addition to the varietal, seasonal and agronomical differences, genomics, moisture content [44] and processing conditions of raisins such as the drying method, temperature and time duration [18]. Studying the effects of different drying methods on the TPCs and antioxidant capacity of seedless purple (Black Monukka) and green raisins (Thompson seedless, Sultana), Qin et al. [39] reported that the TPCs of green and purple seedless raisins were 634.3 mg GAE/100 g and 536.8 mg GAE/100 g for room drying, respectively. They showed that room drying preserved the phytochemical contents of raisins better than sun or shielding film drying. Using different solvents for the extraction of some antioxidants from dried fruits such as prunes, apricots, raisins and figs, Ouchemoukh et al. [45] reported that raisins were rich in TPC (1.18 g GAE/100 g db). Zemni et al. [46] determined the effect of two drying processes (hot air drying in a convective oven and greenhouse drying) and chemical pre-treatments on the quality characteristics of raisins, and compared the characteristics of raisins with sun-dried samples. The highest polyphenol content was 417.7 mg GAE/100 g db in sun-dried samples, while the lowest was 207.9 mg GAE/100 g db in oven-dried samples. In our study, the range of TPCs in raisins was 332.7 to 659.6 mg GAE/g db, which is higher than the range reported by Qin et al. [39], Meng et al. [41], Ouchemoukh et al. [45] and Zemni et al. [46] but is in good agreement with the range reported by Mnari et al. [42].

Raisins are commonly consumed as a snack food around the world because of their delicious taste and consumer acceptability. The sensory properties of dried fruits such as raisins, especially their consumer acceptability, are usually critical since some raisins may contain seeds. In our study, panellists evaluated raisins according to their colour, appearance, odour, taste, hardness, chewiness and overall likability (Table A5). The sensory colour scores for the raisins of the Kay Gray and Mars Seedless cultivars were significantly low in comparison to those of the Kishmish Vatkana, Rhea, Philipp, Campbell Early, Glenora and Özer Beyazı cultivars. The sensory properties of raisins can be influenced by several factors including varietal differences in grapes, their seed status, drying and processing conditions and the chemical properties of raisins. Khiari et al. [18] evaluated the effect of drying at three temperatures on the physico-chemical, textural, phytochemical and sensory properties of Italian raisins pre-treated using different methods. They reported a firmer texture in raisins obtained by drying 70 °C after a traditional pre-treatment, and they noted that Italian raisins showed 'the characters of sweet, firm, sticky, with strong aroma of dried plum/red berries, average aroma of fig/hay/tobacco and a light aroma

of caramel/vanilla with a slight acidity' in a sensory panel. In our study, we were able to compare the sensory likability scores of raisins from nine different cultivars that were produced using the same processing conditions, and we noted that the chewiness score for Kishmish Vatkana, Rhea and Glenora raisins was the highest. Based on the overall likability scores, the raisins of the Rhea and Kishmish Vatkana cultivars were determined as the most liked ($p < 0.05$).

## 5. Conclusions

In this study, nine different grape cultivars which are resistant/tolerant to fungal diseases and can be grown especially in humid regions with very light pesticide application were selected, and the phytochemical and quality characteristics of their fresh berries and their raisins produced by an air-pump dryer at 45 °C with an air velocity rate of 3 m/s to a final moisture content of about 16% were determined. In recent years, the awareness of consumers of human and environmental health and their demand for nutritious and healthy snacks such as raisins have increased dramatically, and as a result, the cultivars of grapes that require less pesticide have gained importance. Humid regions are not very suitable in terms of both fungal diseases and sun drying due to their ecology. In the context of this study, it was concluded that some grape cultivars and their raisins can provide alternatives for regions where viticulture is very limited. The results of this study indicated that the phytochemical and some quality characteristics of raisins obtained from different grape cultivars (belonging to different *Vitis* species) using an energy-saving heat pump dryer may hold potential for raisin production. Having higher bioactive contents and sensory likability scores, grape cultivars such seedless Rhea and Kishmish Vatkana, which can be grown in humid regions, might hold great potential for raisin production.

**Author Contributions:** Methodology, A.A., Z.G., C.T. and Y.Y.; data curation, A.A. and Z.G.; writing—original draft preparation, A.A. and Y.Y.; conceptualization, A.A. and Z.G.; investigation, Z.G. and A.A.; resources, A.A.; supervision, Y.Y.; project administration, Y.Y.; writing—review and editing, A.A. and Y.Y.; software, A.A. and Z.G.; formal analysis, A.A., C.T. and Z.G.; funding acquisition, C.T.; validation, Z.G., C.T. and A.A. All authors have read and agreed to the published version of the manuscript.

**Funding:** This work was supported by a University of Yalova Scientific Research Project (Grant Number 2020/AP/0001). The sponsors had no role in study design; in the collection, analysis and interpretation of data; in the writing of the report; and in the decision to submit the article for publication.

**Data Availability Statement:** The numerical data and outputs produced in this study are available from the corresponding author upon request.

**Acknowledgments:** The authors are grateful for the support of the Yalova Atatürk Horticultural Central Research Institute, Turkey, where the grapes were grown and the analyses were performed.

**Conflicts of Interest:** The authors declare that there are no conflicts of interest regarding the publication of this paper.

## Appendix A

**Table A1.** Bioactive contents of fresh berries from different grapes cultivars on the basis of fresh and dry weight.

| Cultivar | Total Phenolic Content * (g GAE/100 g) | | Antioxidant Activity by DPPH Assay (mmol TE/100 g) | | Antioxidant Activity by FRAP Assay (mmol TE/100 g) | | Antioxidant Activity by CUPRAC Assay (mg Trolox/100 g) | |
|---|---|---|---|---|---|---|---|---|
| | Fresh Weight ** | Dry Weight | Fresh Weight | Dry Weight | Fresh Weight | Dry Weight | Fresh Weight | Dry Weight |
| Glenora | 8.95 ± 0.50 [d] | 38.36 ± 2.48 [c] | 646.52 ± 6.07 [a] | 2770.55 ± 44.67 [e] | 1320.00 ± 8.33 [cde] | 5657.32 ± 133.38 [cd] | 6.97 ± 0.41 [cd] | 29.87 ± 2.24 [c] |
| Kay Gray | 9.12 ± 0.27 [d] | 39.25 ± 0.75 [c] | 659.65 ± 1.16 [a] | 2839.95 ± 38.17 [ed] | 1388.06 ± 40.04 [cd] | 5976.14 ± 200.24 [cd] | 7.66 ± 0.55 [cd] | 32.97 ± 2.51 [c] |
| Mars Seedless | 10.63 ± 0.05 [c] | 49.94 ± 1.67 [b] | 660.66 ± 0.44 [a] | 3102.10 ± 93.89 [bdc] | 1556.11 ± 33.42 [c] | 7308.82 ± 344.60 [b] | 8.99 ± 0.14 [cd] | 42.17 ± 0.77 [c] |
| Muscat Bleu | 12.75 ± 0.17 [ab] | 65.96 ± 4.94 [a] | 657.63 ± 3.06 [a] | 3404.75 ± 290.99 [ab] | 2922.78 ± 141.81 [a] | 15,114.33 ± 1147.35 [ab] | 292.97 ± 33.70 [ab] | 1526.00 ± 301.22 [ab] |
| Özer Beyazı | 9.16 ± 0.27 [d] | 49.14 ± 0.83 [b] | 661.16 ± 1.58 [a] | 3547.21 ± 72.40 [a] | 1332.50 ± 8.33 [cde] | 7149.95 ± 207.14 [c] | 8.60 ± 1.06 [cd] | 46.10 ± 5.45 [c] |
| Philipp | 12.89 ± 0.29 [a] | 64.88 ± 1.48 [a] | 653.84 ± 1.91 [a] | 3290.03 ± 10.48 [abc] | 3120.00 ± 271.41 [a] | 15,699.63 ± 1369.12 [a] | 446.49 ± 178.04 [a] | 2246.54 ± 895.32 [a] |
| Rhea | 7.20 ± 0.23 [e] | 33.27 ± 1.87 [c] | 647.52 ± 8.71 [a] | 2993.68 ± 90.27 [cde] | 1076.94 ± 15.78 [e] | 4982.02 ± 264.01 [d] | 4.35 ± 0.19 [d] | 20.11 ± 0.61 [c] |
| Kish. Vatkana | 7.52 ± 0.40 [e] | 33.40 ± 2.42 [c] | 525.81 ± 17.97 [b] | 2332.24 ± 37.08 [f] | 1138.05 ± 44.16 [de] | 5053.54 ± 318.66 [d] | 4.75 ± 0.30 [cd] | 21.12 ± 1.79 [c] |
| Campbell Early | 11.93 ± 0.40 [b] | 64.53 ± 4.00 [a] | 657.88 ± 1.32 [a] | 3556.59 ± 94.27 [a] | 2458.89 ± 65.13 [b] | 13,293.41 ± 504.84 [b] | 177.83 ± 14.65 [bc] | 961.27 ± 80.22 [b] |

* Different superscripts within a column indicate significant differences at $p \leq 0.05$. ** All values are expressed as means ± standard deviations ($n = 3$).

**Table A2.** Bioactive contents of raisins from different grape cultivars.

| Cultivar | Total Phenolic Content * (g GAE/100 g) | | Antioxidant Activity by DPPH Assay (mmol TE/100 g) | | Antioxidant Activity by FRAP Assay (mmol TE/100 g) | | Antioxidant Activity by CUPRAC Assay (mg Trolox/100 g) | |
|---|---|---|---|---|---|---|---|---|
| | Fresh Weight ** | Dry Weight | Fresh Weight | Dry Weight | Fresh Weight | Dry Weight | Fresh Weight | Dry Weight |
| Glenora | 13.09 ± 0.34 [ab] | 15.00 ± 0.40 [e] | 660.91 ± 3.30 [a] | 757.43 ± 2.59 [d] | 2974.17 ± 23.20 [bc] | 3408.61 ± 37.70 [c] | 702.36 ± 87.94 [cd] | 804.70 ± 98.07 [cde] |
| Kay Gray | 12.76 ± 0.14 [b] | 15.57 ± 0.19 [de] | 652.57 ± 1.52 [ab] | 796.35 ± 2.68 [c] | 2379.71 ± 255.30 [c] | 2904.26 ± 315.57 [c] | 286.57 ± 70.72 [cde] | 349.70 ± 86.25 [cde] |
| Mars Seedless | 13.06 ± 0.18 [ab] | 15.95 ± 0.24 [cd] | 650.56 ± 1.91 [ab] | 794.59 ± 4.13 [c] | 1981.11 ± 30.71 [c] | 2419.79 ± 42.89 [c] | 94.67 ± 14.65 [e] | 115.65 ± 18.04 [e] |
| Muscat Bleu | 13.30 ± 0.07 [ab] | 16.80 ± 0.08 [b] | 653.84 ± 1.16 [ab] | 826.23 ± 4.73 [b] | 4978.33 ± 60.09 [ab] | 6290.81 ± 72.82 [ab] | 2128.83 ± 197.65 [b] | 2690.38 ± 254.92 [b] |
| Özer Beyazı | 12.82 ± 0.22 [b] | 19.30 ± 0.32 [a] | 658.39 ± 5.69 [a] | 991.01 ± 20.98 [a] | 2383.89 ± 237.18 [c] | 3589.82 ± 386.55 [c] | 580.82 ± 52.84 [cde] | 875.27 ± 95.23 [cd] |
| Philipp | 13.58 ± 0.21 [a] | 15.39 ± 0.17 [de] | 638.18 ± 0.76 [c] | 723.60 ± 2.14 [e] | 6962.50 ± 2414.84 [a] | 7893.51 ± 2733.81 [a] | 4505.21 ± 571.24 [a] | 5109.87 ± 666.23 [a] |
| Rhea | 13.40 ± 0.14 [a] | 15.64 ± 0.12 [de] | 659.14 ± 0.87 [a] | 769.01 ± 1.73 [d] | 3189.44 ± 194.73 [bc] | 3270.69 ± 218.08 [bc] | 369.73 ± 99.87 [cde] | 431.23 ± 115.82 [cde] |
| Kis. Vatkana | 13.05 ± 0.23 [ab] | 15.45 ± 0.32 [de] | 654.60 ± 1.75 [ab] | 775.33 ± 4.99 [cd] | 2481.11 ± 230.50 [c] | 2938.15 ± 264.74 [c] | 219.41 ± 14.65 [de] | 259.88 ± 17.51 [de] |
| Campbell Early | 13.32 ± 0.16 [ab] | 16.48 ± 0.20 [bc] | 644.75 ± 8.78 [bc] | 797.30 ± 10.82 [c] | 3518.61 ± 79.31 [bc] | 4351.28 ± 106.72 [bc] | 833.49 ± 116.07 [c] | 1030.93 ± 145.72 [c] |

* Different superscripts within a column indicate significant differences at $p \leq 0.05$. ** All values are expressed as means ± standard deviations ($n = 3$).

**Table A3.** Some of the physico-chemical properties of fresh grapes and their raisins from different cultivars.

| Cultivar | Moisture * (%) | | Water Activity | | Soluble Solids Content (°Brix) | |
|---|---|---|---|---|---|---|
| | Fresh ** | Raisin | Fresh | Raisin | Fresh | Raisin |
| Glenora | 76.66 ± 0.48 [d] | 12.74 ± 0.31 [ef] | 0.86 ± 0.01 [ab] | 0.43 ± 0.01 [ef] | 18.30 ± 1.13 [bc] | 54.90 ± 0.20 [bc] |
| Kay Gray | 76.77 ± 0.29 [d] | 18.05 ± 0.13 [c] | 0.85 ± 0.00 [ab] | 0.42 ± 0.01 [f] | 17.23 ± 0.38 [bc] | 51.70 ± 0.10 [cde] |
| Mars Seedless | 78.69 ± 0.64 [bcd] | 18.13 ± 0.19 [c] | 0.85 ± 0.02 [ab] | 0.46 ± 0.00 [ed] | 18.60 ± 0.92 [b] | 55.80 ± 0.36 [b] |
| Muscat Bleu | 80.60 ± 1.56 [ab] | 20.86 ± 0.32 [b] | 0.86 ± 0.01 [ab] | 0.65 ± 0.01 [b] | 16.80 ± 0.46 [bc] | 50.40 ± 0.46 [de] |
| Özer Beyazı | 81.36 ± 0.43 [a] | 33.54 ± 1.79 [a] | 0.85 ± 0.01 [ab] | 0.71 ± 0.02 [a] | 17.47 ± 0.15 [bc] | 52.40 ± 0.21 [bcd] |
| Philipp | 80.13 ± 0.01 [abc] | 11.80 ± 0.37 [f] | 0.87 ± 0.01 [a] | 0.49 ± 0.00 [d] | 16.23 ± 0.15 [c] | 48.70 ± 0.32 [e] |
| Rhea | 78.35 ± 0.82 [cd] | 14.29 ± 0.24 [de] | 0.84 ± 0.02 [b] | 0.45 ± 0.01 [edf] | 17.13 ± 0.25 [bc] | 51.46 ± 0.10 [cde] |
| Kishmish Vatkana | 77.46 ± 0.58 [d] | 15.57 ± 0.32 [d] | 0.85 ± 0.01 [ab] | 0.63 ± 0.02 [b] | 21.70 ± 1.60 [a] | 65.10 ± 0.45 [a] |
| Campbell Early | 81.49 ± 0.52 [a] | 19.13 ± 0.21 [bc] | 0.85 ± 0.01 [ab] | 0.57 ± 0.02 [c] | 17.97 ± 0.42 [bc] | 53.90 ± 0.10 [bcd] |

* Different superscripts within a column indicate significant differences at $p \leq 0.05$. ** All values are expressed as means ± standard deviations ($n = 3$).

**Table A4.** Colour properties (CIELab) of fresh grapes and their raisins from different cultivars and colour difference values after drying.

| Cultivar | CIE L * | | CIE a * | | CIE b * | | Chroma | | Hue Angle | | Colour Difference (ΔE *) |
|---|---|---|---|---|---|---|---|---|---|---|---|
| | Fresh ** | Raisin | Fresh | Raisin | Fresh | Raisin | Fresh | Raisin | Fresh | Raisin | |
| Glenora | 53.50 ± 2.30 [a] | 51.95 ± 1.90 [a] | 0.39 ± 0.02 [c] | 1.36 ± 0.41 [b] | −0.01 ± 0.92 [bc] | −1.97 ± 0.08 [c] | 0.81 ± 0.33 [d] | 2.41 ± 0.19 [bc] | −0.14 ± 1.19 [ab] | −0.97 ± 0.15 [e] | 2.70 ± 0.96 [c] |
| Kay Gray | 55.51 ± 2.05 [a] | 39.45 ± 3.20 [b] | −3.32 ± 0.53 [e] | 1.04 ± 0.65 [bc] | 0.18 ± 1.40 [bc] | 4.60 ± 0.86 [a] | 3.52 ± 0.43 [b] | 4.73 ± 0.97 [a] | −0.09 ± 0.41 [ab] | 1.36 ± 0.11 [a] | 17.22 ± 1.31 [a] |
| Mars Sdls. | 41.67 ± 2.40 [bc] | 51.45 ± 1.10 [a] | 0.97 ± 0.44 [bc] | 0.23 ± 0.08 [c] | −0.99 ± 0.18 [cd] | −3.84 ± 0.54 [d] | 1.44 ± 0.18 [cd] | 3.84 ± 0.54 [ab] | −0.82 ± 0.31 [b] | −1.51 ± 0.01 [f] | 10.24 ± 1.90 [b] |
| Muscat Bleu | 35.33 ± 1.35 [c] | 50.90 ± 1.04 [a] | 1.29 ± 0.24 [bc] | 0.29 ± 0.04 [c] | −1.32 ± 0.33 [cd] | −4.37 ± 0.50 [d] | 1.87 ± 0.10 [cd] | 4.38 ± 0.50 [a] | −0.79 ± 0.21 [ab] | −1.50 ± 0.02 [f] | 15.91 ± 2.32 [a] |
| Özer Beyazı | 42.38 ± 2.43 [bc] | 38.14 ± 2.70 [bc] | −1.56 ± 0.47 [d] | 2.67 ± 0.49 [a] | 4.42 ± 1.15 [a] | 3.39 ± 1.38 [a] | 4.74 ± 0.91 [a] | 4.42 ± 0.84 [a] | −1.21 ± 0.18 [b] | 0.87 ± 0.27 [bc] | 6.08 ± 0.83 [bc] |
| Philipp | 38.63 ± 4.08 [bc] | 32.90 ± 2.03 [c] | 0.95 ± 0.23 [bc] | 0.28 ± 0.07 [c] | −1.63 ± 0.54 [cd] | −0.12 ± 0.04 [b] | 1.92 ± 0.33 [cd] | 0.31 ± 0.05 [d] | −1.01 ± 0.26 [b] | −0.43 ± 0.18 [d] | 6.03 ± 1.85 [bc] |
| Rhea | 45.54 ± 2.17 [b] | 42.23 ± 1.00 [b] | 3.60 ± 0.41 [a] | 2.41 ± 0.11 [a] | 1.91 ± 0.21 [b] | 4.47 ± 0.46 [a] | 4.09 ± 0.26 [ab] | 5.08 ± 0.38 [a] | 0.49 ± 0.09 [a] | 1.07 ± 0.05 [ab] | 4.62 ± 2.27 [c] |
| Kis. Vatkana | 35.62 ± 2.89 [c] | 40.85 ± 0.98 [b] | 1.72 ± 0.13 [b] | 0.54 ± 0.04 [bc] | −2.76 ± 0.53 [d] | −1.03 ± 0.13 [bc] | 3.26 ± 0.46 [b] | 1.16 ± 0.09 [cd] | −1.01 ± 0.09 [b] | −1.08 ± 0.08 [e] | 5.66 ± 1.87 [bc] |
| Cam. Early | 37.16 ± 2.30 [c] | 40.54 ± 1.41 [b] | 1.29 ± 0.35 [bc] | 0.60 ± 0.02 [bc] | −1.54 ± 0.2 [cd] | 0.34 ± 0.05 [b] | 2.03 ± 0.07 [c] | 0.69 ± 0.04 [d] | −0.88 ± 0.20 [b] | 0.52 ± 0.05 [c] | 3.96 ± 0.77 [c] |

* Different superscripts within a column indicate significant differences at $p \leq 0.05$. ** All values are expressed as means ± standard deviations ($n = 3$).

**Table A5.** Results of sensory evaluation of the raisins of different grape cultivars on a scale from 1 (dislike very much) to 10 (like very much).

| Cultivar | Colour * | Appearance ** | Odour | Taste | Hardness | Chewiness | Overall Likability |
|---|---|---|---|---|---|---|---|
| Glenora | 7.70 ± 0.68 cd | 8.30 ± 0.68 a | 7.90 ± 1.10 bc | 9.10 ± 0.57 a | 8.90 ± 0.57 a | 9.10 ± 0.78 a | 8.50 ± 0.53 b |
| Kay Gray | 6.50 ± 0.71 e | 6.20 ± 0.79 d | 6.10 ± 0.74 f | 5.90 ± 1.52 c | 6.40 ± 1.27 cd | 5.80 ± 1.14 d | 6.30 ± 0.95 e |
| Mars Seedless | 6.50 ± 0.53 e | 5.90 ± 0.88 d | 6.80 ± 0.42 ef | 6.70 ± 1.16 bc | 7.80 ± 0.63 b | 8.00 ± 0.94 b | 7.20 ± 0.79 c |
| Muscat Bleu | 7.10 ± 0.99 de | 6.30 ± 1.49 d | 6.80 ± 0.79 ef | 6.50 ± 1.08 bc | 6.00 ± 1.41 cd | 6.20 ± 1.75 cd | 6.40 ± 1.17 de |
| Özer Beyazı | 7.70 ± 0.82 cd | 8.10 ± 0.99 ab | 7.30 ± 0.95 cde | 6.90 ± 1.52 bc | 7.00 ± 1.33 bc | 6.90 ± 1.29 c | 7.40 ± 1.07 c |
| Philipp | 8.20 ± 0.79 bc | 6.60 ± 1.58 cd | 7.10 ± 0.99 de | 6.60 ± 1.27 bc | 5.50 ± 1.90 d | 4.30 ± 1.42 e | 6.20 ± 0.79 e |
| Rhea | 8.80 ± 0.92 ab | 8.80 ± 1.40 a | 8.80 ± 0.92 a | 9.60 ± 1.27 a | 9.10 ± 0.88 a | 9.60 ± 0.52 a | 9.50 ± 0.97 a |
| Kishmish Vatkana | 9.00 ± 0.67 a | 9.00 ± 0.67 a | 8.50 ± 0.71 ab | 9.30 ± 0.82 a | 9.20 ± 0.79 a | 9.60 ± 0.51 a | 9.40 ± 0.52 a |
| Campbell Early | 8.00 ± 0.94 c | 7.30 ± 1.06 bc | 7.60 ± 0.97 cd | 7.20 ± 1.03 b | 6.50 ± 1.08 cd | 6.50 ± 1.18 cd | 7.10 ± 0.88 cd |

* Different superscripts within a column indicate significant differences at $p \leq 0.05$. ** All values are expressed as means ± standard deviations ($n = 10$).

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
