# Peer review of "Comparison of Important Quality Characteristics of Some Fungal Disease Resistance/Tolerance Grapes Dried with Energy-Saving Heat Pump Dryer"

_agronomy, doi:10.3390/agronomy12040909_

Round 1

Reviewer 1 Report

This research compared nutritional quality characteristic (total phenolic contents, antioxidant activity by DPPH, FRAP and CUPRAC) of nine fungal disease resistance/tolerance grapes dried with energy-saving heat pump dryer and carried out the sensory evaluation as well. 

In my opinion, this study is very interesting and meaningful, especially for raisins production.

However, there are few aspects should be further improved. The followings are my feedback and suggestions.

  1. The main issues for this paper are the result section and discussion section. There are limited information in the result section while discussion section contains too much result description. Therefore, my suggestion is rearranging and separating the result description and discussion clearly.
  2. Some basic information of the experiment design was not stated clearly in the abstract, such as the processing methods and the quality characteristics in the sentence “For the first time, alternative processing methods and important quality characteristics of disease resistant/tolerant grape varieties of different species that can be grown in humid regions were investigated in this study (line 20-22)”, therefore the abstract should be further improved.
  3. In the introduction section , please add some necessary information about energy-saving heat pump dryer as well as the advantage and disadvantage of it, because it is the only one main processing method/machine used in the research.
  4. Reasons of choosing parameters “total phenolic content” , “antioxidant activity” as the important quality characteristics should be stated briefly in the introduction section.
  5. In line 63, it mentioned sultani cultivar is the main seedless grape cultivar used in raisin production in Turkey and many countries.  So why sultani cultivar was not included in the research as the quality controlling group?
  6. In line 91-92, it says “the main objective of this study was to determine the potential of these cultivars for the production of raisins.” Aims of the study should be further clearly described.
  7. In line 95-95, details of the climate, weather, year, rainfall and other information for growing included cultivars should be described in the material and method section.
  8. Beside cultivar, specie, berry colour, special flavour and seed status are also clearly presented in Table 1. However, the effect of these parameters are not analysed and presented. It would be more interesting and more comprehensive if these parameters were included in the analyses.
  9. The expire data of raisins dried by the energy-saving heat pump dryer should be assessed as well as one of the important quality characteristics.
  10. DDPH in line 166 Should be DPPH
  11. The correlation between parts publications cited in discussion section and the experiment results are not clearly (including line 321,416). Further discussion and clearly statement are required, for example, summarising the benefit of phenolics and antioxidants in the discussion then proving that higher phenolics and antioxidants, the better quality of raisins is.
  12. In the sensory evaluation, what is the way to get the overall liking score? If there is a modelling  applied to the final overall liking scores, based on  colour, appearance, odour, taste hardness, chewiness.

Author Response

Reviewer 1

Comments and Suggestions for Authors

This research compared nutritional quality characteristic (total phenolic contents, antioxidant activity by DPPH, FRAP and CUPRAC) of nine fungal disease resistance/tolerance grapes dried with energy-saving heat pump dryer and carried out the sensory evaluation as well.

In my opinion, this study is very interesting and meaningful, especially for raisins production.

However, there are few aspects should be further improved. The followings are my feedback and suggestions.

The main issues for this paper are the result section and discussion section. There are limited information in the result section while discussion section contains too much result description. Therefore, my suggestion is rearranging and separating the result description and discussion clearly.

We are very thankful for this valuable comment of the reviewer. We realized that some parts of the results section were deleted accidentally while converting the manuscript into the journal format. We included missing parts of the results section and corrected this section according to the reviewer's suggestions.

Some basic information of the experiment design was not stated clearly in the abstract, such as the processing methods and the quality characteristics in the sentence “For the first time, alternative processing methods and important quality characteristics of disease resistant/tolerant grape varieties of different species that can be grown in humid regions were investigated in this study (line 20-22)”, therefore the abstract should be further improved.

In accordance with the reviewer's recommendation, the relevant sentence was improved and included in the abstract.

In the introduction section, please add some necessary information about energy-saving heat pump dryer as well as the advantage and disadvantage of it, because it is the only one main processing method/machine used in the research.

In accordance with the referee's recommendations, more information about the heat pump dryer used in the study was added both to the introduction and the methods section.

Reasons of choosing parameters “total phenolic content”, “antioxidant activity” as the important quality characteristics should be stated briefly in the introduction section.

An explanation of the importance of total phenolic content and antioxidant activity also why they are accepted as quality components was added to the introduction section with cited literatures.

In line 63, it mentioned Sultani cultivar is the main seedless grape cultivar used in raisin production in Turkey and many countries.  So why Sultani cultivar was not included in the research as the quality controlling group?

Sultani cultivar is known all over the world as Sultanina, Sultani or Sultani seedless. This cultivar is a grape cultivar grown intensively in western Turkey, especially in Manisa and İzmir provinces. This cultivar is grown for table and drying purposes. However, it is a member of the V. vinifera species and is affected by fungal diseases, so it cannot be grown in humid regions in the north of Turkey. The aim of our study was to examine some quality components of both fresh and raisins that can be grown in humid regions. For this reason, we did not include a cultivar that cannot grow in humid regions. In sum, Sultani cultivar was not included in the study because it did not serve our major purpose.

In line 91-92, it says “the main objective of this study was to determine the potential of these cultivars for the production of raisins.” Aims of the study should be further clearly described.

A detailed explanation was added to the last part of the Introduction section upon the request of the reviewer

In line 95-95, details of the climate, weather, year, rainfall and other information for growing included cultivars should be described in the material and method section.

A detailed explanation was added to the 2.1 Grape Cultivars section upon the request of the reviewer

Beside cultivar, specie, berry colour, special flavour and seed status are also clearly presented in Table 1. However, the effect of these parameters are not analysed and presented. It would be more interesting and more comprehensive if these parameters were included in the analyses.

In fact, since the characteristics of the cultivars directly affect the results of our study, comments about these characteristics were made especially in the discussion section. In particular, the presence of seeds and berry colour affected the content of phenolic compounds, while in the sensory evaluations of dried grapes, “seedlessness” significantly influenced sensory scores as a significant parameter. Briefly, especially in the discussion part of the manuscript, data on many characteristics of the cultivars were included in the discussion and interpreted adequately.

The expire data of raisins dried by the energy-saving heat pump dryer should be assessed as well as one of the important quality characteristics.

Since the water activity and moisture content of raisins were in a safe range, no further study was planned nor carried out during storage. It is known from previous studies that the storage period of dried products with a low water activity value is about a year. For this reason, no experiment/research was made regarding the storage period in our study.

DDPH in line 166 Should be DPPH

Corrected as it was recommended in the revised manuscript.

The correlation between parts publications cited in discussion section and the experiment results are not clearly (including line 321,416). Further discussion and clearly statement are required, for example, summarising the benefit of phenolics and antioxidants in the discussion then proving that higher phenolics and antioxidants, the better quality of raisins is.

In the literature, correlation coefficients among bioactive contents of raisins determined by different assays are available but we did not run the correlation analysis among the parameters studied in our study even though we could do it if requested. The following sentence was re-paraphrased in the conclusion section as the reviewer suggested:

Having higher bioactive contents and sensory liking scores, and grape cultivars like seedless Rhea and Kishmish Vatkana, which can be grown in humid regions, might have a great potential for raisin production.

In the sensory evaluation, what is the way to get the overall liking score? If there is a modelling applied to the final overall liking scores, based on colour, appearance, odour, taste hardness, chewiness.

In fact, there are some explanations about how this evaluation was assessed in the 2.8 Sensory Evaluation section. We strongly believe that clarification is necessary for this section. We re-paraphrased and clarified this section in the following paragraph:

Experienced panellists (5 males and 5 females in the range of 30–50 years old, staff of the AHCRI (Yalova Atatürk Horticultural Central Research Institute, Yalova, Turkey) participated in sensory evaluation sessions. Prior to testing, panellists received a semi-training session related to the seven sensory attributes of raisins such as colour, appearance (such as colour homogeneity, raisin size, raisin size homogeneity and size of wrinkles), odour, taste (such as sweet, sour, bitter and astringent), hardness, chewiness (textural properties) and overall liking so that each panellist could thoroughly discuss and clarify the attributes of samples to be evaluated [24]. All trials were assessed at a consistent condition of controlled temperature, adequate illumination and free from odour. Drinking water and crackers were provided between samples to cleanse the palate. After evaluating each sample, panellists were asked to assign a numerical value between 1 (dislike very much) and 10 (like very much) for raisins.

Reviewer 2 Report

The study examined the total phenolic content, antioxidant activity (DPPH, FRAP and CUPRAC assays), moisture water activity, soluble solids content, color of nine fresh grape cultivars and raisins produced from these cultivars. The study also evaluated the sensory properties of raisins produced from these nine grape cultivars. The research is noteworthy since it examined species that can be grown in humid regions and also used an energy-saving heat pump dryer. The photos of the fresh grapes and the raisins produced from the cultivars were informative.

Good details were provided on the experimental procedures.  It would be useful to get more details on the heat pump dryer.

I think that the discussion section could be condensed as there is extensive discussion of previous grape research

Can you explain why the values for the total phenolic content of the fresh grapes are similar to the values for the corresponding raisins (Tables A1 and A2)? It seems that the raisins should have much higher values than the fresh grapes.

The manuscript is thorough and scientifically sound. 

Page 9, line 256: from 33.27 to 65.96g GAE/100g...

Page 9, line 292: The highest..

Page 11, line 371: for most cultivars.  

Author Response

Reviewer 2

Comments and Suggestions for Authors

The study examined the total phenolic content, antioxidant activity (DPPH, FRAP and CUPRAC assays), moisture water activity, soluble solids content, colour of nine fresh grape cultivars and raisins produced from these cultivars. The study also evaluated the sensory properties of raisins produced from these nine grape cultivars. The research is noteworthy since it examined species that can be grown in humid regions and also used an energy-saving heat pump dryer. The photos of the fresh grapes and the raisins produced from the cultivars were informative.

Good details were provided on the experimental procedures.  It would be useful to get more details on the heat pump dryer.

Detailed information about the energy-saving heat pump used in the study was added to the method section of the manuscript.

I think that the discussion section could be condensed as there is extensive discussion of previous grape research

The discussion section had appeared condensed because part of the result section was omitted accidentally during converting the manuscript into the journal format. The result section was added to the revised manuscript, so all sections should seem more balanced in the revised manuscript.

Can you explain why the values for the total phenolic content of the fresh grapes are similar to the values for the corresponding raisins (Tables A1 and A2)? It seems that the raisins should have much higher values than the fresh grapes.

When the A1 and A2 tables, where the results obtained in the study are given, are carefully examined, it can be seen that the values of fresh grapes on fresh weight basis are lower than those of raisins. All of the values for raisins are higher than the values for fresh berries, without exception. Especially, there is a significant difference among the values in Table A1, which is compatible with similar studies in the literature. Extraction of phenolic compounds from a dried product like raisins is usually difficult in comparison to fresh products, which could explain lower total phenolic contents of raisins in dry matter basis in Table A2 than those in Table A1. Moreover, drying and drying conditions may have a reducing effect on the total phenolic contents of raisins in dry matter basis. It would be more accurate to evaluate both tables independently of each other and not to compare them.

The manuscript is thorough and scientifically sound.

Page 9, line 256: from 33.27 to 65.96g GAE/100g...

Page 9, line 292: The highest..

Page 11, line 371: for most cultivars. 

Corrections were made in the relevant parts of the manuscripts.

Reviewer 3 Report

It is an interesting study and has innovative approach in terms of justifying resistance varieties to be used as a nutritionally valuable snack, while cutting down pesticide residues. Although, several points has to be revised. Objective of the study to propose integration of resistant varieties as a raisin in human diet by a) using non-traditional method of drying and b)comparing their nutritional quality parameters. In this case wouldn't it be more clear to compare just fresh basis of both products (fresh berry and pump-dried raisin), since it is the final product of consumption. It is obvious that dry weight will have higher values than fresh. I haven't notice anywhere it was mentioned why there is need for both fresh and dry weight bases. Overall results and objectives are not very well connected, maybe better to indicate bioactive compound concentrations in supplementary part. See my additional comments below:

line 118-121-It is not clear how many samples were collected and for each cultivar (n=?); what does it mean "at least 10 fruits-do you mean berries?-analyzed in each replication"? means you had 10 replication or as you mentioned 3 replication. Please, revise this part.

Extraction of bioactive compounds- in this paragraph extraction method is described for fresh berries, only in the last sentence there is a mention about raisin. Did you do the same extraction for raisins? Did you grind raisins with blender or grinder? Please, revise this section, because your samples are fresh and dry.

What type of analysis or method you used for sensory analysis? Please, clearly indicate it, before describing.

Discussion part is too long and it is more suits to literature review than discussion. It needs to be reduced and revised in a way to link results to the previously done studies, justify or criticize obtained results.

Author Response

Reviewer 3

Comments and Suggestions for Authors

It is an interesting study and has innovative approach in terms of justifying resistance varieties to be used as a nutritionally valuable snack, while cutting down pesticide residues. Although, several points has to be revised. Objective of the study to propose integration of resistant varieties as a raisin in human diet by a) using non-traditional method of drying and b) comparing their nutritional quality parameters. In this case wouldn't it be more clear to compare just fresh basis of both products (fresh berry and pump-dried raisin), since it is the final product of consumption. It is obvious that dry weight will have higher values than fresh. I haven't notice anywhere it was mentioned why there is need for both fresh and dry weight bases. Overall results and objectives are not very well connected, maybe better to indicate bioactive compound concentrations in supplementary part. See my additional comments below:

We believe that it is not easy to compare the results of phytochemical analyses scientifically in fresh basis since the final moisture contents may have a great influence on the bioactive contents of dried products like raisins. That is why we preferred to present the results in dry matter basis. On the other hand, for nutritional point of view, bioactive contents of dried products in fresh weight basis are more meaningful. These two points are the main reasons of presenting values in both fresh and dry matter basis in our study. Part of the results section was omitted accidentally in the original manuscript during its conversion into the journal format. That might be the main reason of a weak connection between results and objectives. We included missing parts of the results in the revised manuscript, and we hope that it would be satisfactory. Please let us know if we need further clarification. 

line 118-121-It is not clear how many samples were collected and for each cultivar (n=?); what does it mean "at least 10 fruits-do you mean berries?-analyzed in each replication"? means you had 10 replication or as you mentioned 3 replication. Please, revise this part.

More detailed and clear statements about the preparation of grapes for analysis and the replications used in the study have been added under the "2.3. Moisture Content, Water Activity and Brix Values" heading.

Extraction of bioactive compounds- in this paragraph extraction method is described for fresh berries, only in the last sentence there is a mention about raisin. Did you do the same extraction for raisins? Did you grind raisins with blender or grinder? Please, revise this section, because your samples are fresh and dry.

In the extraction of raisins, the same method was used as for fresh grapes, and information about this was added to the relevant part of the manuscript.

What type of analysis or method you used for sensory analysis? Please, clearly indicate it, before describing.

We clarified the sensory analysis in the methods section as stated also by Reviewer 1.

Discussion part is too long and it is more suits to literature review than discussion. It needs to be reduced and revised in a way to link results to the previously done studies, justify or criticize obtained results.

Part of the results section was omitted accidentally in the original manuscript during its conversion into the journal format. We included missing parts of the results in the revised manuscript, and all sections should seem more balanced in the revised version.

Round 2

Reviewer 3 Report

Author fulfilled previously required parts. However, discussion part still needs to be revised!

It is very long, should be reduced. Instead of writing one paragraph of result and long paragraph of literature review, author should integrate/match important aspects of results with literature. In the last paragraph I only see description of result part. 

Author Response

We are reduced the discussion part also tried to summarize the literatures in second time revised manuscript. I hope it is suitable this time. Many thanks for your close cooperation